# RELATIONAL FORWARD MODELS FOR MULTI-AGENT LEARNING

**Andrea Tacchetti\*, H. Francis Song\*, Pedro A. M. Mediano\*, Vinicius Zambaldi,**
**János Kramár, Neil C. Rabinowitz, Thore Graepel, Matthew Botvinick & Peter W. Battaglia**
\* denotes equal contrubtion
Google DeepMind
`{atacchet,songf,pmediano,vzambaldi`
`janosk,ncr,thore,botvinick,peterbattaglia}@google.com`

## ABSTRACT

The behavioral dynamics of multi-agent systems have a rich and orderly structure, which can be leveraged to understand these systems, and to improve how artificial agents learn to operate in them. Here we introduce Relational Forward Models (RFM) for multi-agent learning, networks that can learn to make accurate predictions of agents' future behavior in multi-agent environments. Because these models operate on the discrete entities and relations present in the environment, they produce interpretable intermediate representations which offer insights into what drives agents' behavior, and what events mediate the intensity and valence of social interactions. Furthermore, we show that embedding RFM modules inside agents results in faster learning systems compared to non-augmented baselines. As more and more of the autonomous systems we develop and interact with become multi-agent in nature, developing richer analysis tools for characterizing how and why agents make decisions is increasingly necessary. Moreover, developing artificial agents that quickly and safely learn to coordinate with one another, and with humans in shared environments, is crucial.

## 1 INTRODUCTION

The study of multi-agent systems has received considerable attention in recent years and some of the most advanced autonomous systems in the world today are multi-agent in nature (e.g. assembly lines and warehouse management systems). In particular, research in multi-agent reinforcement learning (MARL), where multiple learning agents perceive and act in a shared environment, has produced impressive results (Jaderberg et al., 2018; Pachocki et al., 2018; Leibo et al., 2017; Hughes et al., 2018; Peysakhovich & Lerer, 2017a; Lerer & Peysakhovich, 2017; Bansal et al., 2017; Lanctot et al., 2017).

One of the outstanding challenges in this domain is how to foster coordinated behavior among learning agents. In hand-engineered multi-agent systems (e.g. assembly lines), it is possible to obtain *coordination by design*, where expert engineers carefully orchestrate each agent's behavior and role in the system. This, however, rules out situations where either humans or artificial learning agents are present in the environment. In learning-based systems, there have been some successes by introducing a centralized controller (D'Andrea, 2012; Foerster et al., 2016; 2017; Hong et al., 2017; Lowe et al., 2017). However, these cannot scale to large number of agents or to mixed human-robot ensembles. There is thus an increasing focus on multi-agent systems that learn how to coordinate on their own (Jaderberg et al., 2018; Pachocki et al., 2018; Perolat et al., 2017).

Alongside the challenges of learning coordinated behaviors, there are also the challenges of measuring them. In learning-based systems, the analysis tools currently available to researchers focus on the functioning of each single agent, and are ill-equipped to characterize systems of diverse agents as a whole. Moreover, there has been little development of tools for measuring the contextual interdependence of agents' behaviors in complex environment, which will be valuable for identifying the conditions under which agents are successfully coordinating.

Here we address these two challenges by developing Relational Forward Models (RFM) for multi-agent systems. We build on recent advances in neural networks that effectively perform relational reasoning with graph networks (GN) (Battaglia et al., 2018) to construct models that learn to predict the forward dynamics of multi-agent systems. First, we show that our models can surpass previous top methods on this task (Kipf et al., 2018; Hoshen, 2017). Perhaps more importantly, they produce intermediate representations that support the social analysis of multi-agent systems: we use our models to propose a new way to characterize what drives each agent's behavior, track when agents influence each other, and identify which factors in the environment mediate the presence and valence of social interactions. Finally, we embed our models inside agents and use them to augment the host agent's observations with predictions of others' behavior. Our results show that this leads to agents that learn to coordinate with one another faster than non-augmented baselines.

## 1.1 RELATED WORK

Relational reasoning has received considerable attention in recent years and researchers have developed deep learning models that operate on graphs, rather than vectors or images, and structure their computations accordingly. These methods have been successfully applied to learning the forward dynamics of systems comprised of multiple entities and a rich relational structure, like physics simulation, multi-object scenes, visual question answering and motion-capture data (Scarselli et al., 2009; Battaglia et al., 2016; Raposo et al., 2017; Santoro et al., 2017; Gilmer et al., 2017; Watters et al., 2017; Kipf et al., 2018; Zambaldi et al., 2018). Recently, this class of methods have been shown to successfully predict the forward dynamics of multi-agent systems, like basketball or soccer games, and to some extent, to provide insights into the relational and social structures present in the data (Hoshen, 2017; Kipf et al., 2018; Zhan et al., 2018; Zheng et al., 2017).

With the recent renaissance of deep-learning methods in general, and of deep reinforcement learning (RL) in particular, considerable attention has been devoted to developing analysis tools that allow researchers to understand what drives agents' behavior, what are the most common failure modes and provide insights into the inner workings of learning systems (Zeiler & Fergus, 2013; Yosinski et al., 2015; Olah et al., 2017; Morcos et al., 2018). In particular, Rabinowitz et al. (2018) embed entire behavioral trajectories of single RL agents as points in an unstructured, high-dimensional space, relying on the inherent structure of the data to yield interpretable representations of whole behavioral motifs.

Finally, coordination in multi-agent system has been a topic of major interest as of late and some of the most advanced MARL systems rely on the emergence of coordination among teammates to complete the task at hand (Jaderberg et al., 2018; Pachocki et al., 2018). Despite these recent successes, coordination is still considered a hard problem and several attempts have been made to promote the emergence of coordination by relaxing some assumptions (Foerster et al., 2016; 2017; Sukhbaatar et al., 2016; Raileanu et al., 2018; He et al., 2016a). Here we show that, by embedding RFM modules in RL agents, they can learn to coordinate with one another faster than baseline agents, analogous to imagination-augmented agents in single-agent RL settings (Hamrick et al., 2017; Pascanu et al., 2017; Weber et al., 2017).

## 2 RELATIONAL ANALYSIS OF MARL SYSTEMS

### 2.1 METHODS

Our RFM is based on graph networks (GN) (Battaglia et al., 2018), and is trained by supervised learning to predict the dynamics of multi-agent systems. Our model takes as input a semantic description of the state of the environment, and outputs either an action prediction for each agent, or a prediction of the cumulative reward each agent will receive until the end of the episode. We show that our model performs well at these tasks, and crucially, produces interpretable intermediate representations that are useful both as analysis tools, and as inputs to artificial agents who can exploit these predictions to improve their decision-making.

Figure 1: (a) The RFM module stacks a GN Encoder, a Graph GRU and a GN Decoder to obtain a relational reasoning module that holds state information across time steps. (b) Example of an environment graph representation. Edges connect agents (magenta and orange) to all entities and are color-coded according to the identity of the receiver. (c) RFM-augmented agents, the output of the the RFM module is appended to the original observation input to the policy network. The on-board RFM module is trained with full supervision and alongside the policy network.

### 2.1.1 RELATIONAL FORWARD MODELS AND BASELINES ARCHITECTURES

A GN is a neural network that operates on graphs. The input to a GN is a directed graph, $(u, V, E)$, where $u \in \mathbb{R}^{d_u}$ is a graph-level attribute vector (e.g. the score in a football game), $V = \{v_i\}_{i=1:n_v}$ is a set of vertices (e.g. the players) with attributes $v_i \in \mathbb{R}^{d_v}$ (e.g the players' $(x,y)$ coordinates on the pitch), and $E = \{(e_k, r_k, s_k)\}_{k=1:n_e}$ is a set of directed edges which connect sender vertex, $v_{s_k}$ to receiver vertex $v_{r_k}$ and have attribute $e_k \in \mathbb{R}^{d_e}$ (e.g. same team or opponent). The output of a GN is also a graph, with the same connectivity structure as the input graph (i.e., same number of vertices and edges, as well as same sender and receiver for each edge), but updated global, vertex, and edge attributes. See Fig. 1b for an example graph.

The sequence of computations in a GN proceed by updating the edge attributes, followed by the vertex attributes, and finally the global attributes. These computations are implemented via three "update" functions (the $\phi$s) and three "aggregation" functions (the $\rho$s),

$$
\begin{aligned}
e'_k &= \phi^e\left(e_k, v_{r_k}, v_{s_k}, u\right), & \bar{e}'_i &= \rho^{e \to v}\left(E'_i\right), \\
v'_i &= \phi^v\left(\bar{e}'_i, v_i, u\right), & \bar{v}' &= \rho^{v \to u}\left(V'\right), \\
u' &= \phi^u\left(\bar{e}', \bar{v}', u\right), & \bar{e}' &= \rho^{e \to u}\left(E'\right)
\end{aligned}
\tag{1}
$$

where $E'_i = \{(e'_k, r_k, s_k)\}_{r_k=i}$. The edges are updated by $\phi^e$, as a function of the sender vertex, receiver vertex, edge, and global attributes. We term the updated edge attribute the "message", $e'_k$. Next, each vertex, $i$, is updated by aggregating the $e'_k$ messages for which $i = r_k$, and computing the updated vertex attribute, $v'_i$, as a function ($\phi^v$) of these aggregated messages, as well as the current vertex and global attributes. Finally, the global attributes are updated, by $\phi^u$, as a function of the current global attribute and all aggregated $e'_k$ and $v'_i$ attributes.

Since a GN takes as input a graph and outputs a graph, GN blocks can be composed to form more complex, and powerful, architectures. These architecture can also be made recurrent in time by introducing a *state graph*, and using recurrent neural networks (RNNs) as the $\phi$ functions. GN-based architectures can be optimized with respect to some objective function by gradient descent (using backpropagation through time for recurrent implementations). Here we focus on supervised learning using datasets of input-output pairs. See (Battaglia et al., 2018) for further details.

We construct our RFM architecture by arranging three GN blocks as in Fig. 1a. We selected this specific architecture to allow our model to perform relational reasoning steps both on the raw input data, before time recurrence is included, and then again on the output of our time recurrent block. This allows the recurrent block to construct memories of the relations between entities and not simply of their current state.

Architecture details are as follows: input graphs $G^t_{in}$ go through a GN encoder block, a basic GN module whose $\phi^v$, $\phi^e$ and $\phi^u$ are three separate 64-unit MLPs, with 1 hidden layer, and ReLU activations and whose $\rho$ functions are summations. The output of the GN encoder block is used, in conjunction with a *state graph* $G^{t-1}_{hid}$, in a "GraphGRU", where each $\phi$ function is a Gated Recurrent Unit (GRU) (Cho et al., 2014) with a hidden state size of 32 for each of vertices, edges and globals. The GraphGRU's output is then copied into a *state graph* and an *output graph*. The *state graph* is

used in the following time step, while the *output graph* is passed through a GN decoder block. This last block's structure has an identical to the GN encoder's, and outputs the model's predictions (e.g. the actions of each agent).

We compared the prediction performance of our RFM module to two state-of-the-art relational reasoning baselines: Neural Relational Inference networks (Kipf et al., 2018) and Vertex Attention Interaction Networks (Hoshen, 2017). These architectures are similar to our RFM module. In particular, NRI models operate on graph structured data and, with the exception that the graph connectivity map is not given, but rather estimated from trajectories using an auto-encoder architecture, they are identical to our model. VAIN networks are essentially single feed-forward GN blocks where the $\phi^e$ and $\rho^{e \rightarrow v}$ functions take particular and restricted form: $\phi^e(e_k, v_{r_k}, v_{s_k}, u) = e^{\|a(v_{r_k}) - a(v_{s_k})\|^2}$ and $\rho^{e \rightarrow v}(E'_i) = v_i \sum_{s_k} e'_k$, with $a(\cdot)$ a learnable function.

We also compared our full RFM against ablated variants, which allowed us to measure the importance of the relational reasoning component, and of time recurrence. In particular we considered a *Feedforward* model, which had no GraphGRU block, and a *No-relation* model, which was a full fledged RFM module but operated on graphs with only self-connections (i.e., edges where $s_i = r_i$). Finally, we included a vector-based MLP + LSTM model among our baselines, so as to highlight the advantage of using graphs over vector based modules. This last model operated on the concatenation of the vertex attributes and had a standard Encoder MLP (64-units), LSTM (32-hidden units), Decoder MLP (2 hidden layers, 32-units each) architecture. We matched all models for capacity (with the exception of NRI which has about 3x more parameters than other models because of its autoencoder connectivity map estimator). Models were within 3% of each other in terms of number of parameters (as reported by the TensorFlow checkpoint loader).

### 2.1.2 MARL ENVIRONMENTS AND AGENT ARCHITECTURE

We considered three multi-agent environments for our study: *Cooperative Navigation* (Lowe et al., 2017), *Coin Game* (Raileanu et al., 2018) and *Stag Hunt* (Peysakhovich & Lerer, 2017b).

*Cooperative Navigation* (Lowe et al., 2017). Two agents navigate an empty $6 \times 6$ arena to cover two tiles. A reward of $+1$ is given to both agents whenever both tiles are covered, i.e., when each agent is on a tile of its own. Episodes are of fixed length (20 environment steps), to encourage a swift resolution of the underlying assignment problem. The positions of both tiles and the starting positions of each agent are randomized at the start of each episode.

*Coin Game* (Raileanu et al., 2018). Two agents roam an $8 \times 8$ arena populated with 12 coins, 4 of each of 3 colors, for 10 environment steps. Agents can collect coins by stepping on them; out of the 3 coin colors, two colors carried a reward and one a punishment. Crucially, each of the two agents only has access to information about 1 good color. The short episode duration incentivizes agents to quickly infer what the unknown good color is by observing their teammate actions, so that all good coins can be collected. At the end of each episode both agents are rewarded according to how many good coins have been collected by either agent. Conversely, they are penalized according to the number of bad coins collected, again by either agent. The role of each color, coin positions, and starting coordinates for the agents are randomized in each episode.

*Stag Hunt* (Peysakhovich & Lerer, 2017b). We implemented a Markov version of the classic Stag Hunt game where two (or four) agents navigate an arena populated with 3 red Stags (each of which is static, and occupies a $2 \times 2$ tile) and 12 green apples, for 32 environment steps. Agents can collect apples by themselves for a reward of $+1$ or, by both stepping on the same stag, capture it for a reward of $+10$. Collected apples and captured stags became unavailable for some time (denoted by dimmed colors), and at each time step have a small probability of becoming available again. All entities' locations are randomized at the start of each episode.

We trained populations of RL agents to convergence on these three tasks using a multi-agent implementation of importance-weighted actor-learner (Jaderberg et al., 2018; Espeholt et al., 2018), a batched advantage actor-critic (A2C) algorithm. For each episode, a group of agents were randomly sampled, with replacement, from a population of 4 learners; at each time step agents received an ego-centric, top-down view of the environment which was large enough to contain the entire arena, and, in the *Coin Game*, one of the 2 good colors. Agents then selected one of 5 actions to be performed (move left, move right, move up, move down, and stay). Within each agent, the input

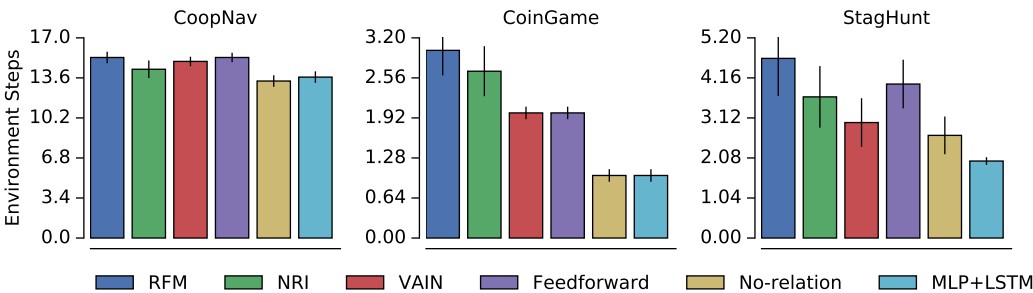

Figure 2: Action prediction performance of our RFM module and baseline models. The reported quantity is the mean number of environment steps for which the predicted actions matched the ground truth exactly, for all agents. Mean across 128 episodes, bars indicate standard deviation across episodes. Alternative measures of model performance show similar results (Fig. 10).

image was parsed by a single convolutional layer ($3 \times 3$-filters, 6 output channels) whose output was fed to a 256-unit single-layer MLP. The MLP output vector was concatenated with each player's last reward, and one-hot encoded last action, as well as, for the *Coin Game*, one of the two coin colors that carried a positive reward. The resulting vector served as input to a 256-hidden-units LSTM whose output was fed into a single soft-max layer. Throughout the learning process, there was no sharing of weights, gradients or any communication channel between the agents, consistent with standard MARL settings.

### 2.1.3 OFFLINE TRAJECTORIES COLLECTION AND MODEL TRAINING

To train the forward RFM model, we collected 500,000 episodes of behavioral trajectories of trained agents acting in their respective environments. At each time step, we collected a semantic description of the state of the environment, as well as the action taken by each agent and the reward they received. These descriptions were compiled into a graph, where agents and static entities (i.e., apples, stags, coins, and tiles) were represented by vertices whose attributes, $v_i$, were: the entity's position in the arena; the one-hot encoded type of the entity (e.g. agent, apple, etc.); (when applicable) the entity's state (e.g. available / collected); and (when applicable) the last action taken. When attributes were not applicable (e.g. the last action of an apple), we padded the corresponding attribute features with zeros. Edges connected all non-agent entities to all agents as well as agents to each other. Input edges contained no attributes and were characterized by senders and receivers only (see Fig. 1b for an example environment graph). In order to understand our analysis contributions, it is crucial to note that while the input graph to our RFM module contained no edge attributes, and edges were simply characterized by their sender and receiver vertices, the edges of a RFM's output graph did contain attributes. These attributes were computed by the network itself and amounted to distributed representations of the effect the sender entity had on the receiver agent.

We also collected 2,500 further episode trajectories for performance reporting and analysis. Training of both RFM and baseline models was conducted using gradient descent to minimize the cross-entropy loss between predicted and ground-truth actions. The training procedure was halted after one million steps, during each of which the gradient was estimated using a batch of 128 episodes. Results are presented in Sec. 2.2.1.

## 2.2 RESULTS

### 2.2.1 ACTION PREDICTION PERFORMANCE

We trained our RFM modules and baseline models to predict the actions of each agent in each of the three games we considered. Models were given a graph representation of the state of the environment, and produced an action prediction for each agent. After training (see Sec. 2.1.3), we used held-out episodes to assess the performance of each model in terms of mean length of perfect roll-out: the mean number of steps during which prediction and ground truth do not diverge. This metric gives

us a measure of how long we could simulate the agents' behavior before making a mistake. For completeness, we report next-action classification accuracy in Sec. A.5.

Results are shown in Fig. 2. As expected, all models achieve similar scores on the *Coop Nav* game, which is a rather simple environment. Our RFM module outperforms the NRI baseline by a substantial margin on the *Coin Game* and *Stag Hunt* environments. Since the two models are identical, except for the initial graph structure inference step, this result suggests that when the importance of some relations is revealed over time, rather than obvious from the start, the graph structure inference step proposed in NRI might not be appropriate. Our RFM consistently outperforms the VAIN model, and on *Stag Hunt* our Feedforward model does as well. This indicates that, for this particular task, distributed interaction representations are superior to simple attention weights. Finally, the MLP+LSTM and No-relation models performed worst across the board, which suggests that relations between entities, rather than the state of the entities themselves, carry most of the predictive power in these environments.

These results reproduce and advance the conclusion that relational models can be trained to perform action prediction for multi-agent systems, and are superior to non-relational models for this task (Kipf et al., 2018; Hoshen, 2017).

### 2.2.2 RELATIONAL ANALYSIS OF THE STAG HUNT GAME: ACTIONS

Here we introduce our relational analysis tools and use the *Stag Hunt* game as a case study. While we illustrate our findings on a simple game, these intuitions can be easily transferred to more complex domains.

We propose the Euclidean norm of a message vector (i.e., $\|e'_k\|$) as a measure of the influence a sender entity, $v_{s_k}$, has on a receiver, $v_{r_k}$. We validate this suggestion in Fig. 3 (top row), where we show that the edge norm between a sender entity (either a stag or an apple) and a receiver agent is predictive of which entity the agent will move towards, or away from, at the next time step.

This intuition can be developed to discover the events that qualitatively change agents' behavior, as well as the factors that mediate how agents interact with one another. Fig. 3 (middle row), for example, shows how the norm of an edge between a stag and an agent changes over time. The importance of the relation is modulated by the prey's state: when a stag becomes available, the edge norm rises substantially; when a stag is consumed, the edge norm drops. Remarkably, the presence or absence of a stag also influences the edge norm between the two teammates, as shown in Fig. 3 (bottom row): in the time step immediately before they consume a stag, the edge between the two teammates is higher than immediately afterwards. In contrast, this effect does not occur with apples, which do not require coordination between teammates to consume. Finally, as shown in Fig. 3 (bottom row), we find that agents' influence on each other's behavior is higher when there is a scarcity of apples (as agents compete for this resource). We note that while significant changes in edge norm or the rank order of edge norm can be used to discover events that qualitatively change agents behavior and factors that mediate agents' social interaction, the raw values have no intrinsic meaning.

Taken as a whole, these findings highlight how the norm of the edge messages, computed by a RFM which is trained to predict the future actions in a multi-agent system, contain intepretable and quantifiable information about when and how certain entities and relations influence agents' behavior, and about which entities and situations mediate the social influence between agents.

### 2.2.3 RELATIONAL ANALYSIS OF THE STAG HUNT GAME: RETURN

A second key finding is that beyond measuring the intensity of a social influence relation, RFM modules can also be used to quantify their *valence*. We trained a RFM model to predict the return received by each agent (until the end of the episode), rather than their future action. We used this model to measure the marginal utility of the actual social context, i.e., to ask: what would happen to agent 1's return if we didn't know the exact state of agent 2?

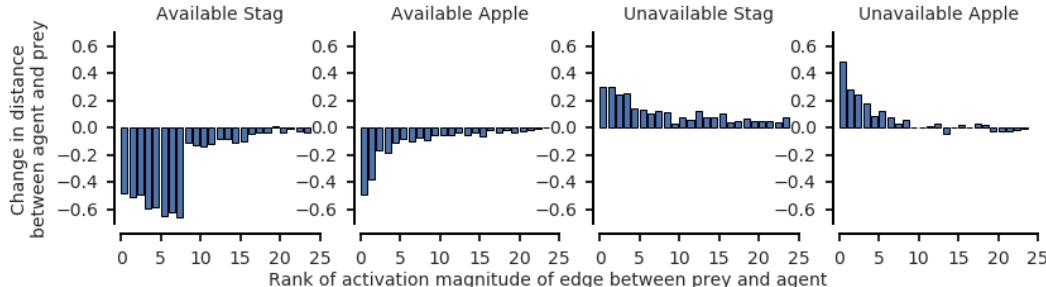

(a) Edge activation magnitude is predictive of future behavior.

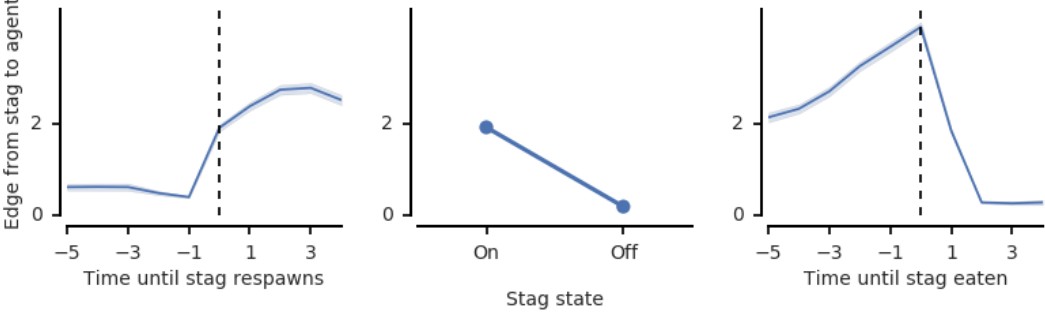

(b) Edge activation magnitude reveals changes in what drives agents behavior over time.

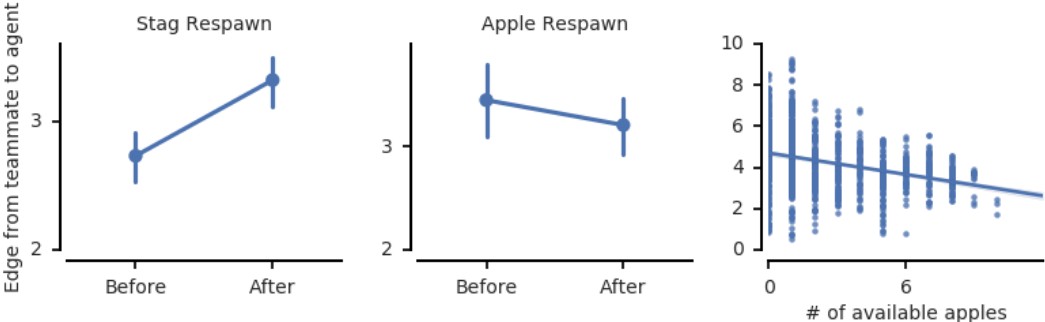

(c) Edge activation magnitude discovers situations that alter agents' social influence.

Figure 3: Edge analysis. Top-row: the norm of output edge activations is predictive of future behavior. On the $y$-axis we plot the average relative displacement between the agent (receiver) and an entity (sender), we order the plot by the rank of the edge activation magnitude; predictive power declines sharply with rank. Middle-row: edge activations discover what agents care about and how this changes over time, here we have time series plots (left and right) of an edge activation norm when a stag becomes available and unavailable, and averages over all time steps grouped by stag state (middle). Bottom row: when stags become available, agents care about each other more than just before that happens, ($p < 0.05$; middle). Apples becoming available has no effect ($p = 0.27$; middle). See also control experiments in Fig. 9. The norm of the edge connecting the two agents is also modulated by scarcity (right), agents compete for apple consumption and the fewer apple there are, the more the two agents influence each other behavior ($r = -0.39$, $p < 0.05$).

The proposed approach is to effectively compare two estimators for agent 1's return:

$$\hat{R}^{a_1}_{\text{Full graph}} = M(s_{a_1}, s_{a_2}, z) \tag{2}$$

$$\hat{R}^{a_1}_{\text{Pruned graph}} = M(s_{a_1}, z) \tag{3}$$

$$\approx \int M(s_{a_1}, s'_{a_2}, z) p(s'_{a_2}|s_{a_1}, z) ds'_{a_2}$$

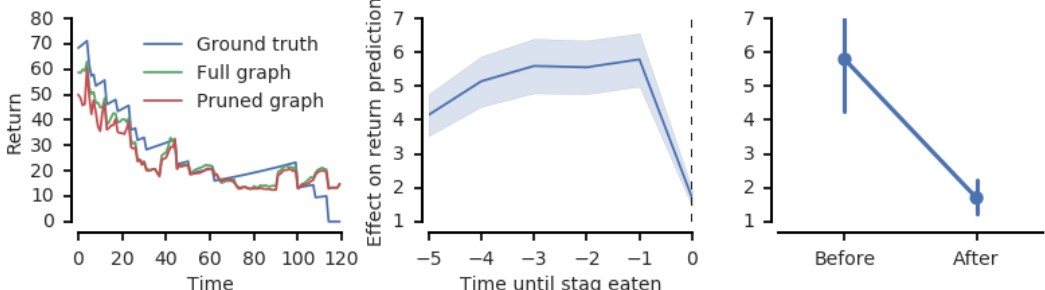

Figure 4: Return analysis: we trained our RFM model to predict the return (until the end of the episode) received by each agent. We trained our model on graphs with and without edges connecting the two agents. (Left) ground truth and predicted return (using both graphs) for a sample episode. (Middle) $\hat{R}^{a_1}_{\text{Full graph}} - \hat{R}^{a_1}_{\text{Pruned Graph}}$ around the time a stag is captured. Positive value indicates that the model estimates that the social influence has a positive marginal utility. (Right) $\hat{R}^{a_1}_{\text{Full graph}} - \hat{R}^{a_1}_{\text{Pruned Graph}}$ right before and right after a stag is captured: agents' influence are most beneficial for each other when they a capture a stag. Episodes ran for 128 steps for this analysis.

where $\hat{R}^{a_1}_{\text{Full graph}}$ is the model $M$'s estimate of the return received by agent 1, given the state of both agents 1 and 2, and all other environment variables, $z$, whereas $\hat{R}^{a_1}_{\text{Pruned graph}}$ is that same estimate, without knowledge of the state of agent 2 (i.e., marginalizing out $s_{a_2}$). In practice, this latter estimate can be obtained by removing the edge connecting the two agents from the input graph[1].

If we find that, in certain situations, the predicted return decreases when removing information about agent 2 (i.e., $\hat{R}^{a_1}_{\text{Full graph}} > \hat{R}^{a_1}_{\text{Pruned graph}}$), we would conclude that the actual state of agent 2 results in a better-than-expected return for agent 1, that is, agent 2 is helping agent 1. Conversely, if the predicted return increases we would conclude that agent 2 is hindering agent 1.

We ran this experiment using a set-up identical to the one we used for action prediction, except for three modifications: (1) the target variable and (2) loss function were changed, from cross-entropy between predicted and ground-truth actions, to mean squared error between predicted and true return; and (3) the training set contained an equal proportion of environment graphs with and without edges between teammates. The latter modification ensured that the pruned-graph computations were not out-of-distribution. The ground truth and predicted return (using both the full and pruned graph) for a sample episode are shown in Fig. 4 (left).

We note that within this setup, both the pruned-graph estimator and the full-graph estimator are produced by a single graph neural network. This network is trained to predict agent 1's return both using the full graph (i.e. knowing the actual state of $a_2$) and the pruned graph (i.e. not knowing the actual state of $a_2$). During training we randomly drop out edges between teammates (to ensure that both full graph and pruned graph are in-distribution for $M$). At test time, we then compute the full-graph estimate by using all edges, and the pruned-graph estimator by dropping out edges between teammates.

Similar to the edge-norm relational analysis above, we can find the entities and events that mediate the value of a social interaction. For example, Fig. 4 (middle and right) show the marginal value of a teammate's particular state (i.e. $\hat{R}^{a_1}_{\text{Full graph}} - \hat{R}^{a_1}_{\text{Pruned Graph}}$) over time and around the time of a stag capture. Thus the model estimates that teammates' specific interactions during this time are beneficial to their return.

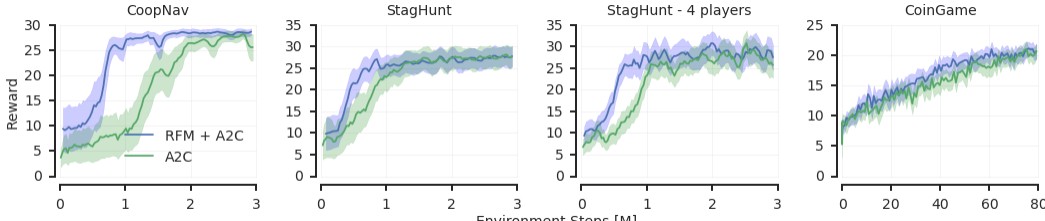

Figure 5: Training curves for A2C agents with and without on-board RFM modules. Allowing agents to access the output of a RFM module results in agents that learn to coordinate faster than baseline agents. This also scales to different number of agents. Importantly, the on-board RFM module is trained alongside the policy network, and there is no sharing of parameters or gradients between the agents. We also show curves for training alongside learning teammates in Fig. 8. Embedding an RFM is also more beneficial than embedding an MLP+LSTM (see Fig. 7.)

## 3 RFM-AUGMENTED AGENTS

### 3.1 METHODS

We have shown that relational reasoning modules capture information about the social dynamics of multi-agent environments. We now detail how these modules' predictions can be useful for improving MARL agents' speed of learning. We extended the agent architecture (described in Sec. 2.1.2) by embedding a RFM module in each agent, and augmenting the policy network's observations with the RFM's output. This agent architecture is depicted in Fig. 1c.

Incorporating an on-board RFM module did not provide the agents with any additional information above and beyond that provided to baseline agents. All games were fully observable, so the additional inputs (i.e. the true last action, and the environment graph, which was provided as input to the embedded RFM) did not add any new information to the original egocentric observations. Similarly, the on-board RFM modules were trained from scratch alongside the policy networks, while the agents were learning to act, so that no additional game structure was given to the augmented agents. Finally, we highlight that each learning agent in the arena had its own RFM module and policy networks; there was never any sharing of weights, gradients or communication between the agents.

Our baseline agent policy network architecture comprised of a CNN that processed the actor's egocentric observation, followed by a MLP+LSTM network that provided action logits (see Sec. 2.1.2 for architecture details). Our augmented agents had an embedded RFM module, which was fed graph representations of the state of the environment, just as in the offline RFM modules in the forward modeling experiments. We trained this module to minimize the cross-entropy loss between its prediction and the last action taken by all fellow agents. We used the prediction output of the on-board RFM module to augment the observation stream at the input of the original policy network. Specifically, the output of the RFM module—predicted action logits for fellow agents—was rendered as image planes whose pixel intensity was proportional to the estimated probability that an agent would be at a certain location at the next time step[2]. These image planes were appended to the ego-centric top-down observation and fed to the original policy network.

### 3.1.1 RESULTS

Our experimental design was relatively straightforward. First, we trained A2C agents (as described in Sec. 2.1.2) to play the three games we considered, as well as a four-player variant of the *Stag Hunt* game. Second, we paired learning agents with these pre-trained experts: learning agents occupied a

---

[1]It is worth highlighting that even though the edge from agent 2 is removed, the estimator $\hat{R}^{a_1}_{\text{Pruned graph}}$ can implicitly take advantage of $s_{a_1}$ and $z$ to effectively form a posterior over $s_{a_2}$ before marginalizing it out. For this reason, we include $p(s_{a_2}|s_{a_1}, z)$ rather than $p(s_{a_2})$ in equation 2.2.3.

[2]For example, consider a fellow agent at the center of the map, and prediction logits indicating that, at the next time step, it might move up with a probability of $0.3$, and down with a probability of $0.7$. The additional image plane would be zero everywhere, with the exception of the pixel above the center (which would have a value of $0.3$) and the one below the center (which would have an value of $0.7$).

single-player slot in each game, while all their teammates were pre-trained experts. We repeated this procedure using both RFM-enhanced agents and baseline A2C agents as learners. During training we recorded the reward received by the singular learning agent in each episode.

Our results show that agents that explicitly model each other using an on-board RFM learn to coordinate with one another faster than baseline agents (Fig. 5). In *Stag Hunt* our RFM-augmented agent achieves a score above 25 after around 600K steps, while baseline agents required around 1M steps. This effect is even more prominent in the 4-player version of the game where these scores are achieved around 500K and 1M steps respectively. Similarly in *Coop Nav* baseline agents required twice as many steps of experience to consistently score above 25 as our RFM-augmented agents. Moreover, in the *Coin Game* environment, the faster learning rate of RFM-augmented agents appears to be due to a superior efficiency in learning to interpret the teammate's action and infer the negative coin color in each episode (see Sec. A.1). Finally, we found that augmenting agents with on-board RFM modules was more beneficial to agents learning than using MLP + LSTM models (see Sec. A.2). These results suggest that agents take into account the on-board RFM's predictions when planning their next action, and that this results in agents that learn faster to coordinate with others, and to discover others' preferences from their actions.

## 4 CONCLUSIONS

Here we showed that our Relational Forward Model can capture the rich social dynamics of multi-agent environments, that its intermediate representations contained valuable interpretable information, and that providing this information to learning agents results in faster learning system.

The analysis tools we introduced allow researchers to answer new questions, which are specifically tailored to multi-agent systems, such as what entities, relations and social interactions drive agents' behaviors, and what environment events or behavior patterns mediate these social and non-social influence signals. Importantly our methods require no access to agents internals, only to behavioral trajectories, making them amenable to analyzing human behavior, sports and ecological systems.

Providing agents with access the output of RFM modules results in agents that learn to coordinate with one another faster than non-augmented baselines. We posit that explicit modeling of teammates and opponents is an important research direction in multi-agent RL, and one that might alleviate the need for communication, parameter sharing or centralized controllers to achieve coordination.

Future work will see our methods applied to more complex and varied domains where artificial and non-artificial agents interact and learn in shared environments. We will focus on identifying entire patterns of behavior for in-agent modeling, so as to adapt the host agent policy more efficiently.

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

# A APPENDIX

## A.1 COIN COLLECTION ANALYSIS IN THE COIN GAME

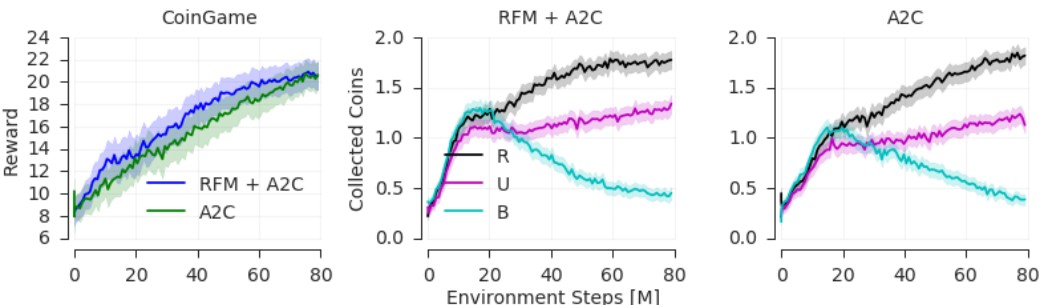

Figure 6: Coin collection analysis in the Coin Game.

As described in the main text, RFM-augmented agents learn the *Coin Game* faster than non-augmented baseline agents. This appears to result from learning more efficiently to discern their teammate's preference. In Fig. 6, the middle panel shows the average number of coins of each color (R: revealed good, U: unrevealed good, B: bad) collected by our RFM-augmented agent during an episode. The right panel shows the same quantities for our baseline agent. We find that the gap between the U curve and the B curve is significantly wider for the RFM-augmented agent than it is for the baseline agent (see, for example, around 50M steps). This suggests that the learning efficiency difference is due to a superior ability to discern the teammate's preferences.

Finally we highlight that our agents, as well as our baselines, vastly outperform previously-published agents on this game: Separate policy predictor agents (He et al., 2016b) and Self-Other Modeling agents (see Fig. 3 in Raileanu et al. (2018). This might imply that the original paper where this game was suggested had poor baseline agents. We suspect this game is not as complex as it may appear, and that baseline agents are close to optimal; this leaves less room for improvement than other games explored in this work.

## A.2 AUGMENTING AGENTS WITH NON-RELATIONAL MODELS

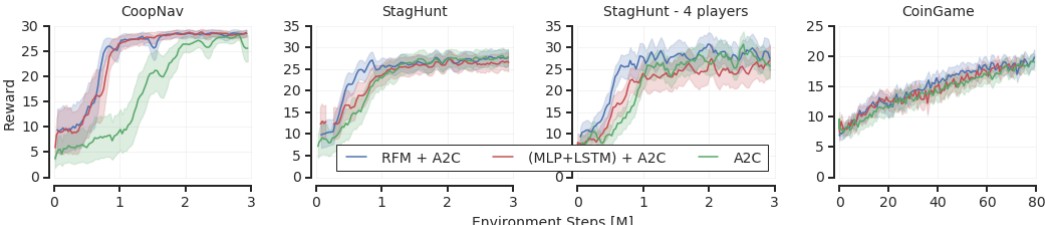

Figure 7: Augmenting agents with predictions from a non-relational model.

In the main text we showed that agents that explicitly model each other using an on-board RFM learn to coordinate with one another faster than baseline agents. For completeness, we show in Fig. 7 the learning performance of agents that have instead been augmented with MLP+LSTM models (as described in the main text). The learning performance of these agents (red) falls between baseline agents (green), which do not explicitly model other agents, and the RFM-augmented agents (blue), which use a relational architecture to model their teammates' behavior. The better performance of RFM-augmented agents is expected, given the more accurate forward predictions that RFMs provide.

## A.3 TRAINING WITH NON-EXPERT TEAMMATES

In the main text we showed how RFM augmented agents learn to coordinate with expert teammates faster than non-augmented baselines. This set-up as is relevant for many interesting situations, e.g.

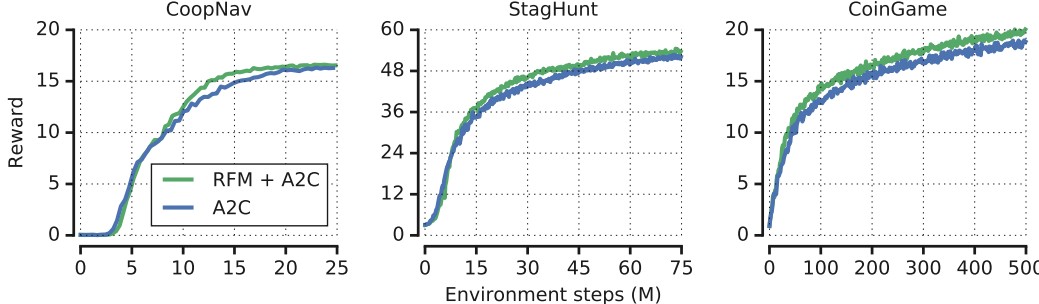

Figure 8: Agents training with non-expert teammates. Reward shown as the average return per agent, averaged over four agent seeds.

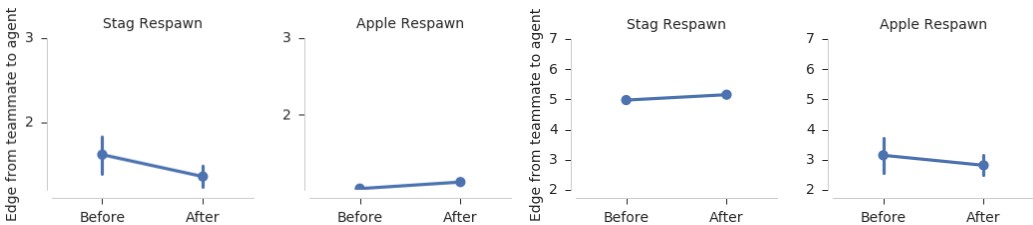

(a) Stags carry no reward. Differences are not signifi-cant ($p = 0.34$ for Stag, $p = 0.46$ for apples).

(b) Stags can be collected by lone hunters. Differences are not significant ($p = 0.67$ for Stag, $p = 0.32$ for apples).

Figure 9: If coordination is not required to collect stags, or if agents are not interested in collecting stags, the edge norm between the two agents is not affected by the appearance of available stags. (Compare to Fig. 3 bottom row left and middle panels.)

when artificial learning agents interact with human experts. For completeness, we show in Fig. 8 the corresponding results when RFM-augments agents train alongside other learning agents (i.e. non-experts). In this case, either all agents in the environment were RFM-augmented (green), or all agents were baseline (blue). We use longer episodes in these experiments (128 steps, rather than 32) in order to make training easier (hence total returns were higher). For brevity, we only report results on the two-player versions of the games.

We see a similar result to the main text: allowing agents to model one another explicitly results in faster learning (e.g. in StagHunt RFM + A2C achieves scores around 48 around 30M steps while vanilla A2C requires 45M training steps. Similarly in CoinGame RFM + A2C achieves a score around 15 in 100M steps while vanilla A2C requires almost 200M steps). We note that this setting presents an additional challenge: a learned model of a teammate's behavior can only provide useful information for coordination after the teammate's policy becomes sensible. The advantage conferred by embedding the RFM into the learning agent will thus be delayed relative to the expert teammate condition shown in the main text. Nonetheless, augmenting agents with RFM models still results in faster learning than omitting it altogether.

## A.4 CONTROL EXPERIMENTS: WHICH EVENTS MEDIATE INTERDEPENDENT BEHAVIOR

In the main text we showed that our RFM model reveals how agents' influence on each other is contextual. In particular, we observe in Fig: 3 (bottom row, left and middle panels) that the Euclidean norm of the activation of the edge between the two agents increases when a stag appears. We argue that this indicates that agents coordinate their behavior when stags are available. Here we report control experiments to test alternative hypotheses. In these experiments, we trained agents on two modifications of the StagHunt game, wherein there is no explicit incentive for agents to coordinate:

- Stags carry no rewards. This tests whether the changes in Fig. 3 could be due to arbitrary changes in the environment. Here our hypothesis predicts that stag appearance would have no effect on edge norms, since agents should learn to ignore them.
- Stags can be collected by a lone hunter. This tests whether the changes in Fig. 3 could be due to the appearance of new reward-carrying objects that do not require coordination. Again, our hypothesis predicts that stags would have no effect on the edge norm between agents, since collecting a stag does not require coordination, like apples.

In both cases our experimental pipeline was as described in the main text: (1) train A2C agents on the modified version of the StagHunt game; (2) collect behavioral trajectories from these agents; (3) train a RFM model to predict the action of each agent given the state of the environment; and (4) report the Euclidean norm of the edge activations in the link connecting the two agents at a time just before and just after a Stag or an Apple appear.

Consistent with our primary hypothesis, there is no significant change in the norm of the activations in the edge connecting two agents when Stags (or apples) appear in these situation. This provides evidence that our RFM model is able to reveal which events in the environment mediate how agents influence each one another.

## A.5 ACCURACY MEASURE

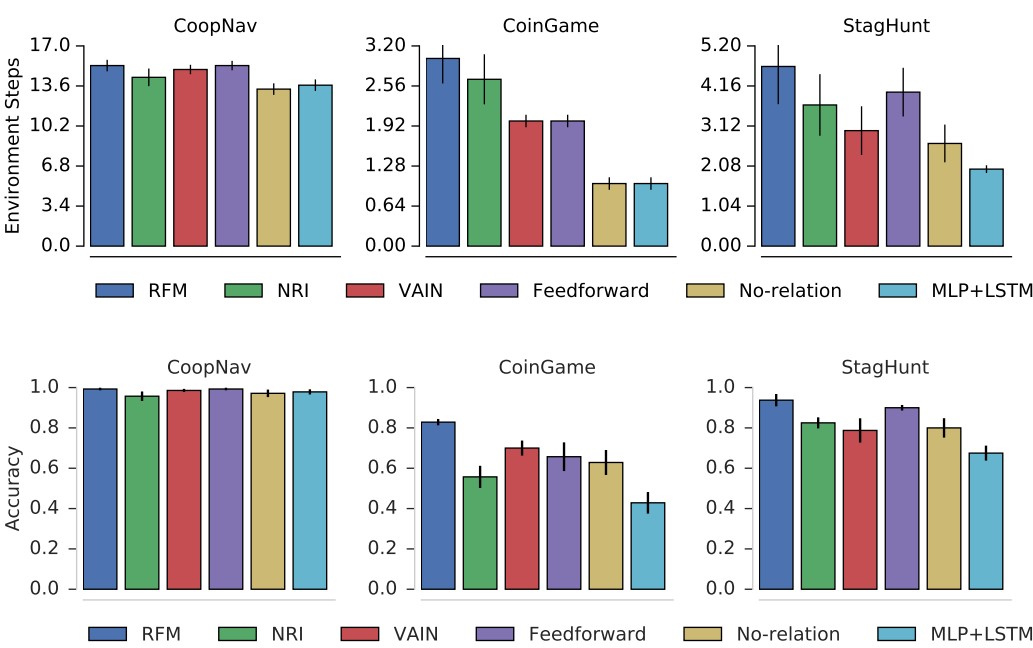

Figure 10: Next-step action classification accuracy.

In the main text we show that our RFM model provides longer perfect rollouts than competing models. Here we provide an alternative metric to measure the relative accuracy of different models. In Fig. 10, we show the next-step action classification accuracy, in the same manner as Fig. 2. Model ranking remain unchanged (with the exception of NRI on CoinGame): the RFM outperforms other models on predicting agent behavior.

