# OpenReview forum: "Relational Forward Models for Multi-Agent Learning"
_ICLR.cc/2019/Conference_

### Official Review · AnonReviewer1 · 2018-11-05

**Rating:** 6
**Confidence:** 4

**Review:**

RELATIONAL FORWARD MODELS FOR MULTI-AGENT LEARNING

Summary: Model free learning is hard, especially in multi-agent systems. The authors consider a way of reducing variance which is to have an explicit model of actions that other agents will take. The model uses a graphical structure and the authors argue it is a) interpretable, b) predicts actions better and further forward than competing models, c) can increase learning speed.

Strong Points:
-	The main innovation here is that the model uses a graph conv net-like architecture which also allows for interpretable outputs of “what is going on” in a game.
-	The authors show that the RFM increases learning speed in several games
-	The authors show that the RFM does somewhat better at forward action prediction than a naïve LSTM+MLP setup and other competing models

Weak Point
-	The RFM is compared to other models in predicting forwards actions but is not compared to other models in Figure 5, so it is not clear that the graphical structure is actually required to speed up learning. I would like to see these experiments added before we can say that the RFM is adding to performance.
-	Related: The authors argue that an advantage of the RFM is that it is interpretable, but I thought a main argument of Rabinowitz et. al. was that simple forward models similar to the LSTM+MLP here were also interpretable? If the RFM does not improve learning above and beyond the LSTM+MLP then the argument comes down to more accurate action prediction (ok) and more interpretability (maybe) which is less compelling.

Clarifying Questions
-	How does the 4 player Stag Hunt work? Do all 4 agents have to step on the Stag together or just 2 of them? How are rewards distributed? Is there a negative payoff for Hunting the stag alone as in the Peysakhovich & Lerer paper?
-	Related: In the Stag Hunt there are multiple equilibria, either agents learn to get plants (which is safe but low payoff) or they learn to Hunt (which is risky but high payoff). Is the RFM leading to more convergence to the Hunting state or is it simple leading agents to learn the safe but low payoff strategies faster?
- The choice of metric in Figure 2 (# exactly correct prediction) is non-standard (not saying it is wrong). I think it would be good to also see a plot of a more standard metric such as loglikelihood of the model's for each of X possible steps ahead. It would help to clarify where the RFM is doing better (is it better at any horizon or is it just able to look further forward more accurately than the competitors?)

---

> ### Author Response · Authors · 2018-11-13
> **Response to AnonReviewer1**
>
> Thank you for your insightful questions and suggestions on the submission.
>
> We hope we have addressed your questions below. In particular, we have addressed your two major weak points: the graphical structure of the embedded RFM is indeed helpful for speeding up learning; and the type and degree of interpretability differs substantially from previous work. We hope you will revise your rating accordingly.
>
>
> 1)  Can we augment agents with models other than the RFM (e.g. MLP + LSTM)?
>
> Yes! We have performed your suggested experiments, which we show in Figure 7. Indeed, the RFM-augmented agents outperform the MLP+LSTM-augmented agents (i.e. those with forward models that are non-relational).
>
> 2)  How does interpretability compare with ToMNet (Rabinowitz et al, 2018)?
>
> This is a good question. To begin, interpretability is not a binary property of a model; there are a number of advantages that RFMs offer over the ToMNet construction in Rabinowitz et al, 2018. Most importantly, the ToMNet embeds sequences of behavior as points in an unstructured, high-dimensional Euclidean space, relying on the inherent structure of the data (and optionally an information bottleneck) to yield interpretable representations. In contrast, RFMs shape behavioral representations through the structure of entities and relations (via the graph net). These representations are very natural for humans to interface with, as they conform to representation schemas of human cognition (Spelke & Kinzler, 2007). Moreover, the representations of the RFM also allow us to easily ask directed questions about how different entities influence agent behavior (e.g. Figure 3), which is not something that the ToMnet enables.
>
> We also note that the ToMNet was designed as a single-agent forward model, while RFMs naturally scale to the multi-agent setting. This allows us to augment agents with the RFM module, which is not something pursued in this previous work.
>
> 3)  Clarifications about Stag Hunt
>
> 4 players - do all 4 need to step on the stag to capture it? No, only 2
> Is there a negative reward for hunting alone? No
> Does including the RFM leads to more Stags being captured? We have only made qualitative observations, but anecdotally, the RFM-augmented agents go for stags like maniacs.
>
> 4)  Choice of metric in Figure 2.
>
> We have now provided an alternative metric in Figure 10 (next-step action classification accuracy) which shows the same qualitative results.
>
> The reason we use the particular metric in the main text is that is gives us a measure of how long the model remains useful. In particular, we are learning a simulator of the agents dynamics; the metric gives an indication of how many steps one can simulate before making a mistake. Alternative rollout metrics are hard to define beyond the first mistake when the true environment and the predictions have diverged. There most likely isn’t a perfect metric that covers all bases, and in particular alternative rollout metrics are hard to define after the model makes a mistake, since the ground-truth observations and predictions no longer match. Nonetheless, between this and the new Figure 10 we believe there’s a strong case that the RFM-based model is better.
>
> References:
> Spelke & Kinzler. (2007). Core knowledge. Developmental science, 10(1), 89-96.

---

> ### Author Response · Authors · 2018-11-23
> **Response to AnonReviewer1 -- 2**
>
> Hello,
>
> Thank you again for taking the time to review for ICLR and for your insightful feedback.
>
> Following your suggestion, we have updated the text to include a more thorough discussion of Rabinowitz et al 2018 in the Related Work section. We highlighted that Rabinowitz et al.’s ToM net focuses on single agent RL and on entire behavioral motifs as opposed to an entity-relation interpretable model of each action and event. Similarly, we pointed to Fig. 7 more prominently in the text; this figure contains an additional experiment showing that onboard RFM modules accelerates agents learning to a larger extent than a non-relational MLP + LSTM based module. Finally, we included our thinking for our choice of model-performance metric in the main text and directed the reader to Fig. 10 where, for completeness, we report the next-action classification accuracy of each model.
>
> We think these additions will help the reader put our work in the context of existing methods, appreciate the problems in which relational models might be a preferred choice, as well as provide more complete performance assessment measures.
>
> With only two days left in the rebuttal period we wanted to make sure we have dispelled your concerns. Please do let us know if anything else needs to be further clarified.
>
> Thank you

---

### Official Review · AnonReviewer3 · 2018-11-07
**Relational Forward Models for Multi-Agent Learning provides a new tool for assessing coordination in MARL and can improve MARL training speeds.**

**Rating:** 7
**Confidence:** 3

**Review:**


This paper used graph neural networks to do relational reasoning of multi-agent systems to predict the actions and returns of MARL agents that they call Relational Forward Modeling. They used RFM to analyze and assess the coordination between agents in three different multi-agent environments. They then constructed an RFM-aumented RL agent and showed improved training speeds over non relational reasoning baseline methods.

I think the overall approach is interesting and a novel way to address the growing concern of how to access coordination between agents in multi-agent systems. I also like how they authors immediately incorporated the relational reasoning approach to improve the training of the MARL agents.

I wonder how dependent this approach is to the semantic representation of the environment. These semantic descriptions are similar to hand crafted features and thus will require some prior knowledge about the environment or task and will be harder to obtain on more difficult environment and tasks.

Will this approach work on continuous tasks? For example, the continuous state and action space of the predator-prey tasks that use the multi-agent particle environment from OpenAi.

I think one of the biggest selling points from this paper is using this method to assess the coordination/collaboration between agents (i.e. the social influence amongst agents). I would have liked to see
more visualizations or analysis into these learned representations. The bottom row of Figure 3 shows that "when stags become available, agents care about each other more than just before that happens". While this is very interesting and an important result, i think that this allows one to see what features of the environment (including other agents) are important to a particular agents decision making but it doesn't really answer whether the agents are truly coordinated, i.e. whether there are any causal dependencies between agents.

For the RFM augmented agents, I like that you are able to train the policy as well as the RFM simultaneously from scratch, however, it seems that this requires you to only train a single agent in the multi-agent environment. If I understand correctly, for a given multi-agent environment, you first pre-trained A2C agents to play the three MARL games and then you paired one of the pre-trained (expert) agents with the RFM-augmented learning agents during training. This seems to limit the practicality and usability of this method as it requires you to have pre-trained agents that have already solved the task. I would like to know why the authors didn't try to train two (or four) RFM-augmented agents from scratch together. When you use one of the agents as a pre-trained agent, this might make the training of the RFM module a bit easier since you have at least one agent with a fixed policy to predict actions from.  It could be challenging when trying to train both RFM modules on two learning agents as the behaviors of learning agents are changing over time and thus the learning might be unstable.

Overall, i think this is an interesting approach and especially for probing what information drives agents' behaviors. However, I don't see the benefit of the RFM-augmented agent provides. It's clearly shown to learn faster than non RFM-augmented agents (which is good), however, unless I'm mistaken, the RFM-augmented agent requires a pre-trained agent to be able to learn in the first place.

--edit:
The authors have sufficiently addressed my questions and concerns and have performed additional analysis.  My biggest concern of weather or not the RFM-augmented agent was capable of learning without a pre-trained agent has been addressed with additional experiments and analysis (Figure 8).

Based on this, i have adjusted my rating to a 7.

---

> ### Author Response · Authors · 2018-11-13
> **Response to AnonReviewer 3 2/2**
>
> 3)  Are the agents truly coordinated? Can we measure causal influence between agents?
>
> This is a good question. There are many potential definitions of coordination. From a game theoretic perspective, the agents have definitely found a coordinative equilibrium. They coordinate to consume stags, which is reflected in their overall return. Another sense of coordination is whether agents’ behaviors are mutually interdependent in service of a common goal. The RFM analysis in the bottom of Figure 3 demonstrates that it is statistically appropriate to describe the agents’ behavior as mutually interdependent when stags are present. The common goal, of course, is stag consumption. A final sense might be whether there are ground-truth causal influences between agents. We note that the RFM approach we pursue here is not designed to answer causal questions, per se: it is a statistical fit to time-series data, and does not traffic directly with interventions or counterfactuals. We are currently exploring such possibilities in ongoing research. If there’s an additional sense of coordination that you’re interested, please feel free to suggest a specific experiment to falsify the conjecture that we’re picking up on something other than coordination here.
>
> As a further test of whether the agents are coordinated in these ways, we ran two additional experiments (1) where there is no scope for coordination between the agents and (2) when no interdependent behavior is required. For additional experiment (1) We trained deep RL agents on a modified version of Stag Hunt where stags yielded no reward at all (i.e. the only rewards are for consuming apples). We then trained the RFM on rollouts of these agents’ behavior. In contrast to the standard case, we found that the edge norms between agents was not modulated by the appearance of a stag (Figure 9a). In additional experiment (2) we obtained similar results in a version of the environment where stags could be consumed by single agents, without the need for coordination (Figure 9b).
>
> 4)  Does the RFM-augmented agent require pre-trained agents?
>
> Not at all. We only chose to include this experiment previously to isolate the benefit of including the RFM in the augmented agent and because this situation is relevant to artificial agnets learning to act in an environment shared with human experts. However, the RFM-augmented agent can be trained in just the same way alongside learning agents too, with similar benefits. We have included this in Figure 8 in a revised version of the manuscript.
>
> We note that when the RFM-augmented agent is trained alongside teammates which are also learning agents, the relative benefit of the RFM on the agent’s return is smaller in magnitude than when this agent is trained alongside expert teammates. We suspect the reason for this is that the RFM model is initially modeling the behavior of untrained teammates, and there are fewer opportunities for rewarded coordination. Since the teammates are learning at roughly the same rate as the RFM-augmented agent, the RFM only has a chance to provide useful information later on during training.
>
> References:
> Battaglia et al, (2016). Interaction networks for learning about objects, relations and physics. NIPS.
> Santoro, et al. (2017). A simple neural network module for relational reasoning. NIPS.
> Watters, et al. (2017). Visual interaction networks: Learning a physics simulator from video. NIPS.
> Barrett, et al. (2018). Measuring abstract reasoning in neural networks. arXiv:1807.04225.
> Zambaldi, et al. (2018). Relational Deep Reinforcement Learning. arXiv:1806.01830.

---

> ### Author Response · Authors · 2018-11-13
> **Response to AnonReviewer3 1/2**
>
> Thank you for your insightful questions and comments on the submission.
>
> We hope we have addressed your questions below. In particular, we have addressed your major criticism (#4 below): RFM-augmented agents do not require pre-trained agents to be able to learn in the first place. Given that this is not a limitation of the method (and we show experiments demonstrating this), we hope you will revise your rating accordingly.
>
> 1)  Is this approach dependent on having semantic representations of the environment?
>
> Yes, at the moment, the approach we describe is dependent on having such a semantic representation. Learning such representations purely from perceptual input is a field of active research. For example, there has been some success in relational reasoning from pixels (e.g. Santoro et al, 2017; Watters et al, 2017; Barrett et al, 2018; Zambaldi et al, 2018), though little attempt has been made to interrogate these systems to uncover the semantics of the intermediate representations (which is something we leverage for our analysis). We do not try to solve the problems of learning semantic representations from pixels here, but we anticipate that as progress is made in this domain, we will be to transfer it over to build the next generation of models.
>
> 2)  Would this work in continuous settings?
>
> Actually, the methods we use originated in continuous settings. For example, graph nets have been used to model the dynamics of interacting particles (Battaglia et al, 2016). In multi-agent settings, both VAIN (Hoshen, 2017) and NRI (Kipf et al, 2018) have been applied to model behavior in continuous domains (soccer and basketball, respectively). We did not explicitly test the RFM model in continuous domains in this submission, but we have no reason to believe that it would not work.

---

### Official Review · AnonReviewer2 · 2018-11-09
**Review Relational Forward Models for Multi-Agent Learning**

**Rating:** 6
**Confidence:** 3

**Review:**

This paper proposes to use graph neural networks in the scenario of multi-agent reinforcement learning (MARL). It tackles two current challenges, learning coordinated behaviours and measuring such coordination.

At the core of the approach are graph neural networks (a cite to Scarselli 2009 would be reasonable): acting and non-acting entities are represented by a graph (with (binary) edges between acting-acting and acting-nonacting entities) and the graph network produces a graph where these edges are transformed into a vectorial representation, which then can be used by a downstream task, e.g. a policy algorithm (as in this paper) that uses it to coordinate behavour. Because the output of the graph network is a structurally identical graph to the input, it is possible to interpret this output.

The paper is well written, the main ideas are clearly described. I'm uncertain about the novelty of the approach, at least the way the RFM is utilized in the policy is a nice idea (albeit, a-posteriori, sounds straight forward in the context of MARL). Similarly, using the graph output for interpretations is an obvious choice). Nevertheless, showing empirically that the ideas actually work gives the paper a lot of credibility for being a stepping stone in the area of MARL.

---

> ### Author Response · Authors · 2018-11-13
> **Response to AnonReviewer2**
>
> Thank you for your comments.
>
> We take it as a good sign that you see the ideas as obvious a posteriori. To the best of our knowledge, though, no one has actually explored them. When others have modelled the dynamics of multi-agent systems (e.g. NRI, VAIN, ToMnet), they have not attempted to integrate these models into the agents themselves. Conversely, in papers that do put models of opponents in MARL, these models are not relational, and they model goals in agents identical to oneself (e.g. Raileanu et al. 2018), policy representations or Q-values (He et al. 2016), but not future actions. In a similar manner, no one has developed this method to interpret multi-agent behavior to the extent that we do here. The event-based analysis and value-based analyses are novel applications of this framework.
>
> Thank you for pointing out that we were missing a citation to Scarselli, we included it the revised manuscript.

---

### Official Review · AnonReviewer4 · 2018-11-12
**Review of Relational Forward Models for Multi-Agent Learning**

**Rating:** 7
**Confidence:** 3

**Review:**

This paper studies predicting multi-agent behavior using a proposed neural network architecture. The architecture, called a relational forward model (RFM) is the same graph network proposed by Battaglia et al., 2018, but adds a recurrent component. Two tasks are define: predict the next action of each agent, and predict the sum of future rewards. The paper demonstrates that RFMs outperform two baselines and two ablations. The authors also show that edge activation magnitudes are correlated with certain phenomenons (e.g. an agent walking towards an entity, or an entity being “on” or “off”). The authors also show that appending the output of a pre-trained RFM to the state of a policy can help it learn faster.

Overall, this paper presents some interesting ideas and is easy to follow, but the significance of the paper is not clear. The architecture is a rather straightforward extensions of previous work, and using graph networks for predictive modeling in multi-agent settings has been examined in the past, making the technical contributions not particularly novel. Examining the correlation between edge activation magnitudes and certain events is intriguing and perhaps the most novel aspect of this paper, but it is not clear how or why this information would be useful. There a few unsubstantiated claims that are concerning. There are also some odd experimental decisions and results that should be addressed.

For specific comments:

1. Why would using a recurrent network help (i.e. RFM vs Feedforward)? Unless the policies are non-Markovian, the entire prediction problem should Markovian. I suspect that most of the gains are coming from the fact that the RFM method simply has more parameters than the Feedforward method (e.g. it can amortize some of the computation into the recurrent part of the network). Suggestion: train a Feedforward model that has more parameters (with appropriate hyperparameter sweeps) to see if this is the cause. If not, provide some analysis for why “memories of the relations between entities” would be any more beneficial than simply recomputing those relations.
2. The other potential reason that the recurrent method did better is that policy actions are highly correlated (e.g. because agents move in straight lines to locations). If so, then recurrent methods can outperform feedforward methods without having to learn anything about what actually causes policies to move in certain directions. Suggestion: measure the correlation between consecutive actions. If there is non-trivial correlation, than this suggests that RFM does better than Feedforward (which is basically prior work of Battaglia et. al.) is for the wrong reasons.
3. If I understand the evaluation metric correctly, for each rollout, it counts how many steps from the beginning of the rollout match perfectly before the first error occurs. Then it averages this “minimum time to failure” across all evaluation rollouts. If this is correct, why was this evaluation metric chosen? A much more natural metric would be to just compute the average number of errors on a test data-set (and if this is what is actually reported, please update the description to disambiguate the two). The current metric could be very deceptive:  Methods that do very well on states around the initial-state distribution but poorly near the end of trajectories (e.g. perfectly predicts the actions in the first 10 steps, but then resorts to random guessing for the last 99999 time steps) will outperform methods that have lower average error rate (e.g. a model that is correct 50% of the time). Suggestion: change the metrics to average number of errors, or report both, or provide a convincing argument why this metric is meaningful.
4. Unless I misunderstood, the results in Section 2.2.3 seem spurious and the claims seem unsubstantiated. For one, if we look at Equations (1) and (2), when we average over s_a1 and s_a2, they should both give the same average for R_a1. Put another way: the prune graph should (in theory) marginalize out s_a2. On average, its expected output should be the same as the output of the full graph (after marginalizing out s_a1 and s_a2). Obviously, it is possible to find specific rollouts where the full graph has higher value than the prune graph (and it seems Figure 4 does this), but it should equally be possible to find rollouts where the opposite is true. I’m hoping I misunderstood this section, but otherwise this seems to invalidate all the claims made in this section.
5. Even if concern #4 is addressed, the following sentence would still seem false: “This figure shows that teammates’ influence on each other during this time is beneficial to their return.” The figure simply shows predictions of the RFM, and not of the ground truth. Moreover, it’s not clear what “teammates’ influence” actually means.
6. The comparison to NRI seems rather odd, since that method uses strictly less information than RFM.
7. For Section 3, is the RFM module pretrained and then fine-tuned with the new policy? If so, this gives the “RFM + A2C” agent extra information indirectly via the pretrained weights of the RFM module.
8. I’m not sure what to make of the correlation analysis. It is not too surprising that there is some correlation (in fact, it’d be quite an interesting paper if the findings were that there wasn’t a correlation!), and it’s not clear to me how this could be used for debugging, visualizations, etc. If someone wanted to analyze the correlation between two entities and a policy’s action, it seems like they could directly model this correlation.

Some minor comments:
 - In Figure 3C, right, why isn’t the magnitude 0 at time=1? Based on the other plots in Figure 3c, it seems like it should be 0.
 - The month/year in many of the citations seems odd.
 - The use of the word “valence” seems unnecessarily flowery and distracting.

My main concern with this paper is that it is not particularly novel and the contribution seems questionable. I have some concerns over the experimental metric and Section 2.2.3, but even if that is clarified, it is not clear how impactful this paper would be. The use of a recurrent network seems unnecessary, unjustified, and not analyzed. The analysis of correlations is interesting, but not particularly compelling or surprising. And lastly, the RFM-augmented results are not very strong.

--

Edit: After discussing with the authors, I have changed my rating. The authors have adjusted some of the language, which I previously thought overstated the contributions and was misleading. They have added a number of experiments which valid the claim that their method is proposing a reasonable way of measuring collaboration. I also realized that I misunderstood one of the sections, and I encourage the authors to improve the presentation to (1) present the significance of the experiments more clear, (2) not overstate the results, and (3) emphasize the contribution more clearly.

Overall, the paper presents convincing evidence that factors in a graph neural networks do capture some notion of collaboration. I do not feel that the paper is particularly novel, but the experiments are thorough. Furthermore, their experiments show that adding an RFM module to an agent consistently helps (albeit not by much). Given that the multi-agent community is still trying to decide how to best quantify and use metrics for collaboration, I find it difficult to access the long-term impact of this paper. However, given the thoroughness of the experiments and analysis, I suspect that this will be valuable for the community and deserves some visibility.

---

> ### Author Response · Authors · 2018-11-13
> **Response to AnonReviwer4 3/3**
>
> 5)  Even if concern #4 is addressed, the following sentence would still seem false: “This figure shows that teammates’ influence on each other during this time is beneficial to their return.” The figure simply shows predictions of the RFM, and not of the ground truth. Moreover, it’s not clear what “teammates’ influence” actually means.
>
> Good catch. We cannot actually compute the ground truth. We have rephrased this to: “Thus the model estimates that teammates' specific interactions during this time are beneficial to their return.”
>
> 6)  The comparison to NRI seems rather odd, since that method uses strictly less information than RFM.
>
> NRI has access to the same set of information as the RFM. In comparison however, NRI actively discards information: it infers a connectivity map from the past trajectory, and may choose to do inference using any number of edges. This might actually confer advantages if the ground-truth process is indeed sparse and relatively stationary. Overall, NRI is a good method and it has many advantages. Our results nevertheless demonstrate that these design decisions are less well-suited to the patterns of multi-agent interaction in these environments.
>
> 7) For Section 3, is the RFM module pretrained and then fine-tuned with the new policy? If so, this gives the “RFM + A2C” agent extra information indirectly via the pretrained weights of the RFM module.
>
> The on-board RFM is trained *from scratch* alongside the policy network. There is no pre-training.
>
>
> 8)  I’m not sure what to make of the correlation analysis. It is not too surprising that there is some correlation (in fact, it’d be quite an interesting paper if the findings were that there wasn’t a correlation!), and it’s not clear to me how this could be used for debugging, visualizations, etc. If someone wanted to analyze the correlation between two entities and a policy’s action, it seems like they could directly model this correlation.
>
> Measuring the emergence of coordinated behavior in multi-agent systems is an important open problem in this field (see also AnonReviewer3’s comment to this effect). Especially in the case of *learning* systems, assessing whether or not agents are able to coordinate, what drives their behavior, whether they learn to help or hinder one another and how they modify their behavior in response to changes in the environment are all crucial aspects of multi-agent analysis that we struggle to quantify. Here we show that learning relational models of multi-agent systems might be a good place to look.
>
> With respect to the particular suggestion that one could directly model correlations between entities and a policy’s actions, we wish it were that simple! The influence of one agent’s state on another’s behavior can be highly contextual, so one would need to factor in the state to tease apart the appropriate effect. This amounts to fitting a parameterized model of the interaction, which is precisely what we’re doing here.
>
>
>
> 9) Some minor comments
> a. In Figure 3C, right, why isn’t the magnitude 0 at time=1? Based on the other plots in Figure 3c, it seems like it should be 0.
> Agents tend to rapidly move away from fruit or stags they just consumed. The probability of respawning is small (0.05) so there will be nothing there for some time (on average). This means that there is some information that a recently-consumed entity can provide to the agent that just consumed it: when the entity is adjacent to the agent, its chances of respawning are lower than otherwise. This predictive information drops sharply after the first step as agents will move towards available entities (which might be anywhere) rather than directly away from unavailable ones (see also Figure 3 top row the two right panels).
>
> b. The month/year in many of the citations seems odd.
> We have fixed this.
>
> c. The use of the word “valence” seems unnecessarily flowery and distracting.
> “Valence” is a standard technical term from psychology, denoting the attractiveness or aversiveness of a stimulus (e.g. Frijda, 1986).
>
> References
> Frijda, N. H. (1986). The emotions. Cambridge University Press.

---

> > ### Comment · AnonReviewer4 · 2018-11-14
> > **Re:  Response to AnonReviwer4 3/3**
> >
> > 5-7) Thank you for the clarification and update.
> > 8) Perhaps this is a bit subject and still ill-defined for the multi-agent learning community, but it's still not clear to me what "measuring the emergence of coordinated behavior" means or how the analysis provided does this. What may help me (and other readers) understand the significance of the work is to include specific, concrete uses for these correlations.
> >
> > A more grounded concern: The numbers in Figure 3 are now compared to anything. For example, in Figure 3c, left: Is a change in magnitude from 2.7 to 3.2 large? How does this compare to the edge from teammate to apple? From stag to teammate? It's not surprising that when the input changes, the magnitude of some hidden layer of a graph network changes. What would be interesting is if the magnitude of the hidden units only changed when the stag appeared and changed to a much larger degree than any other hidden unit.
> >
> > Regardless, the language around the analysis seems a bit overstated. "Our models explain what drives each agent’s behavior, track how agents influence each other, and what factors in the environment mediate the presence and valence of social interactions." The use of the word "explains" may be why Reviewer #3 is asking about causality, since this model seems to only construct correlations. It's also not clear to me why the analysis shows "how agents influence each other." Nor do I see an explanation for "what factors in the environment mediate...valence of social interactions." It seems the only analysis for this looked at the presence of an edge in the RFM (i.e. prune vs full graph).
> >
> > 9) Okay.

---

> > > ### Author Response · Authors · 2018-11-14
> > > **Re: Response to AnonReviwer4 3/3**
> > >
> > > Thank you for raising these concerns.
> > >
> > > 8a) Learning coordinated behavior is a goal of multi-agent reinforcement learning. Apart from the coarse game-theoretic definition, coordinated behavior is abstractly defined as inter-dependent behavior towards a common objective. Yet this does not give clear direction on how to actually *measure* whether a set of policies acts in a coordinated manner or not. The way we structure the RFM model allows us to do this directly. This can be useful to any researcher that wants to design either tasks or algorithms that lead to coordinated behavior: we now have a way to measure whether they got it or not. In turn, this will assist the MARL community in building better algorithms, and better tasks.
> > >
> > > Here’s a concrete example. Suppose we want to design the next generation of house-cleaning robots. In particular, we’re designing a pair of robots, one of whom operates a dustpan, and one operates a brush. As engineers designing this system, we would want to ensure that the agents are actually learning a coordinated solution to the dusting problem. Our method could allow us to identify whether a particular pair of policies is actually achieving this by fitting a powerful non-linear predictive model that goes well beyond simple correlation and co-occurrence. Moreover, we could use it to find the situations for which the robots’ behavior is most inter-dependent, and use this to design training environments or curricula to increase the learning pressure for such behavior.
> > >
> > > We have added some clarifying description in the introduction of the paper:
> > >
> > > “Alongside the challenges of learning coordinated behaviors, there are also the challenges of measuring them. In learning-based systems, the analysis tools currently available to researchers focus on the functioning of each single agent, and are ill-equipped to characterize systems of diverse agents as a whole. Moreover, there has been little development of tools for measuring the contextual inter-dependence of agents' behaviors in complex environment, which will be valuable for identifying the conditions under which agents are successfully coordinating.”
> > >
> > > 8b) You're correct that the absolute magnitudes (e.g. 2.7 and 3.2) are not meaningful in and of themselves. What matters is the relative values. The change in magnitude is certainly statistically significant, as can be seen from the error bars. The edges from stags to agents change even more than this when the stags respawn, as shown in Figure 3b.
> > >
> > > With respect to your suggestion that it's not surprising that when the input changes, the activation of hidden units changes too: this is not a given! In particular, we now show in Figure 8 that these results do not show up when stags are not relevant for coordination (i.e. their presence does not mediate coordinative behaviors).
> > >
> > > We added a comment in Section 2.2.2 to the effect that the raw numbers are not intrinsically meaningful, and that one should instead consider comparisons between edge norm values or rank order of edge norm values.
> > >
> > > 8c) We agree that "explains" and "how" claim too much ground. We have changed this sentence to "Our models enable a characterization of what drives each agent's behavior, tracking when agents influence each other, and identifying factors in the environment which mediate the presence and valence of social interaction.". With respect to your last point, the analyses in Figure 3 and 9 study the presence, and Figure 4 studies the valence.

---

> > > > ### Comment · AnonReviewer4 · 2018-11-15
> > > > **Re: Response to AnonReviwer4 3/3**
> > > >
> > > > 8a) I see. I feel that this is a bit subjective, but I recognize that the community is still trying to decide how to best measure coordination. To the extent that this is a proposed way of doing that, I see the merit of the work. I think the paper would be much better received if it clearly presented the method as a *proposed* way of measuring coordination (and motivate the experiments as a way of demonstrating why this proxy is a good proxy), rather than presenting it as a method that solves the nebulous "coordination measurement" problem.
> > > >
> > > > 8b) While the relative magnitude between events may be interesting, relative magnitude between neurons would be more insightful, but there's an overarching concern that it is not even clear if hidden unit activation magnitude is meaningful. Figure 9 (9, not 8, right?) does convey this information to a satisfying degree.
> > > >
> > > > I'll end this point by making the following suggestion:
> > > > What we seem to care about is d a_a1 / d s_a2, where a_a1 = action of agent 1 and s_a2 = state of agent 2. If the paper showed that the magnitude of the hidden unit was more correlated with d a_a1 / d s_a2 than the magnitude of any other hidden unit, then this would seem to give even stronger, more direct evidence for the claim that, "the magnitude of this neuron is a good proxy for measuring coordination."
> > > >
> > > > 8c) Great, thank you for the change.

---

> > > > > ### Author Response · Authors · 2018-11-16
> > > > > **Re: Response to AnonReviwer4 3/3**
> > > > >
> > > > > 8a) Thank you for your suggestion, we have changed the paragraph in our introduction that outlines our contribution as follows:
> > > > >
> > > > > [...] Perhaps more importantly, they produce intermediate representations that support the social analysis of multi-agent systems: we use our models to propose a new way to characterize what drives each agent's behavior, track when agents influence each other, and identify which factors in the environment mediate the presence and valence of social interactions. [...]
> > > > >
> > > > > Similarly we modified the motivation of our experiments in Sec. 2.2.2 as follows:
> > > > >
> > > > > We propose the Euclidean norm of a message vector (i.e., $\|e'_k\|$) as a measure of the influence a sender entity, $v_{s_k}$, has on a receiver, $v_{r_k}$. We validate this suggestion in Fig.~\ref{fig:edges} (top row), where [...]
> > > > >
> > > > > 8b) Thank you, we are happy to hear you recognize the value of our work and we are glad you agree that Figure 9 (sorry about the typo!) conveys the information that only relative changes, and not raw values are meaningful.
> > > > >
> > > > > 8c) Thank you for the thoughtful suggestion. We’ve given similar ideas a lot of thought and can share some insights.
> > > > >
> > > > > First, we penalize edge activations during training, precisely to encourage the edges to only convey useful information for prediction. As a result of this, if the state s_a2 is not predictive of the action a_a1, the model will learn to suppress the edge activation. In other words, if the derivative you propose is small, the edge norm should be small.
> > > > >
> > > > > In the ideal case, we would have ground-truth data of the effect of s_a2 on a_a1 to validate this. While we do not have ground-truth data for agent-agent influence (see below), we do have them for object-agent influence (Fig. 3 top-row). In this case, we find that objects (stags and apples) with large edge norms are informative about the direction that the agent subsequently travels, while objects with small edge norms are not informative in the same way. Thus the magnitude of these edges are good proxies for measuring object-agent influence.
> > > > >
> > > > > Being able to validate that the agent-agent edge norm correlates with this derivative turns out to be technically complex, even in these relatively small environments. For one, in the case of the apples and stags above, the ground-truth influence of objects on agents can be estimated through their attractive effect on the agent (i.e. one knows the action to measure correlation against); while the effect of one agent’s state on another agent’s action can be much more intricate (e.g. I’m going for this apple, so you’d better not). Another complexity is that the derivative measure you propose needs to be integrated over plausible alternative values for s_a2. Choosing a space of counterfactuals, or averaging over a proposal distribution q(s_a2), brings its own challenges. We’re actually working on a similar idea in spirit at the moment (though it’s out of the scope of this work), and we look forward to sharing our results in a later paper when they are ready.

---

> ### Author Response · Authors · 2018-11-13
> **Response to AnonReviewer4 2/3**
>
> 3)  If I understand the evaluation metric correctly, for each rollout, it counts how many steps from the beginning of the rollout match perfectly before the first error occurs. Then it averages this “minimum time to failure” across all evaluation rollouts. If this is correct, why was this evaluation metric chosen? A much more natural metric would be to just compute the average number of errors on a test data-set (and if this is what is actually reported, please update the description to disambiguate the two). The current metric could be very deceptive:  Methods that do very well on states around the initial-state distribution but poorly near the end of trajectories (e.g. perfectly predicts the actions in the first 10 steps, but then resorts to random guessing for the last 99999 time steps) will outperform methods that have lower average error rate (e.g. a model that is correct 50% of the time). Suggestion: change the metrics to average number of errors, or report both, or provide a convincing argument why this metric is meaningful.
>
> We have now provided an alternative metric in Figure 10 (next-step action classification accuracy) which shows the same qualitative results.
>
> The reason we use the particular metric in the main text is that is gives us a measure of how long the model remains useful. In particular, we are learning a simulator of the agents dynamics; the metric gives an indication of how many steps one can simulate before making a mistake. There most likely isn’t a perfect metric that covers all bases, and in particular alternative rollout metrics are hard to define after the model makes a mistake, since the ground-truth observations and predictions no longer match. Nonetheless, between this and the new Figure 10 we believe there’s a strong case that the RFM-based model is better.
>
>
> 4)  Unless I misunderstood, the results in Section 2.2.3 seem spurious and the claims seem unsubstantiated. For one, if we look at Equations (1) and (2), when we average over s_a1 and s_a2, they should both give the same average for R_a1. Put another way: the prune graph should (in theory) marginalize out s_a2. On average, its expected output should be the same as the output of the full graph (after marginalizing out s_a1 and s_a2). Obviously, it is possible to find specific rollouts where the full graph has higher value than the prune graph (and it seems Figure 4 does this), but it should equally be possible to find rollouts where the opposite is true. I’m hoping I misunderstood this section, but otherwise this seems to invalidate all the claims made in this section.
>
> We think we’ve identified the cause of the misunderstanding. You’re correct that if one averaged over *all* situations (i.e. if one marginalized out both s_a1 and s_a2, and z as well), then the two estimators should give the same results. In other words, the full-graph estimator and the pruned-graph estimator should have the same mean value. But here we’re not averaging over all situations. We’re picking a particular *class* of situations.
>
> To give an example: in the middle of Figure 4, each point represents a particular distribution of situations q_t(s_a1, s_a2, z). These situations are defined by the time t until stag consumption. These distributions, q_t, are themselves not equal to the overall marginal distribution p(s_a1, s_a2, z). Thus when you compute the expected value of the full-graph estimator under q_t, and the expected value of the pruned-graph estimator under q_t, you get different quantities.
>
> Here’s an analogy: imagine we build two models for John’s heart rate. One model includes more factors than the other model. They agree on the average value though: marginalizing over all circumstances, John’s heart rate averages at 80bpm. The richer model also includes a particular factor: when Mary is in the room, John’s heart rate goes up by 10bpm. If we marginalize over situations when Mary is present, the richer model estimates John’s average heart rate to be 90bpm, while the smaller model estimates it as 80bpm. The two models may still agree in expectation over *all* situations (though, as you intuit, the richer model would have to make up for this somehow by having John’s heart rate being lower on average when Mary is absent).
>
> Overall, this allows us to measure whether an interaction is overall “good” or “bad” for the agents.
>
> (Some extra info if it helps: the left panel of Figure 4 shows the differences from a single episode, for illustration purposes. We compute the quantities in the middle and right panels over 10 randomly-chosen episodes from the test set).

---

> > ### Comment · AnonReviewer4 · 2018-11-14
> > **Re: Response to AnonReviewer4 2/3**
> >
> > 3) Thank you for including this Figure! The update addresses this concern.
> > 4) I think I should clarify my concern, because I don't think I conveyed it clearly. Let's say X is a random variable representing John's heart rate. I understand that if E[X | Mary=1] > E[X], then this suggests that the presence of Mary tends to increase X. However, my point is that the prune graph does not model E[X]; it seems to directly model E[X | Mary=1]. To use the analogy: if you trained the "simple model" on data only when Mary was present, then you would expect the two models to estimate the same BMP when Mary is present.
> >
> > To tie it back to the paper, "Mary being present" is equivalent to "the other agent being present." Unless I'm mistaken, the prune graph is only trained on data collected when the other agent is present, correct? The paper states, "In practice, this latter estimate
> > can be obtained by removing the edge connecting the two agents from the input graph" which to me suggests that the presence of this one edge is the only difference (and so they train on the same data, i.e. when the other agent/Mary is always present).
> >
> > It seems like the only way to actually get a model of E[X] is to get data when Mary isn't there, i.e. collect data when the other agent is not there. It wouldn't be particularly surprising to find that a value function trained on this data would predict lower average values, but if the hypothesis is that another agent's presence seems helpful, it seems necessary to compare when the other agent is present with when the agent is not present.
> >
> > Another (new) question: Why is removing the edge sufficient for removing knowledge about agent 2? Do you mean that all edges from s_a2 are removed? It seems like the RFM still has access to both s_a1 and s_a2 when making the predictions. I apologize for not including this question in the original review.

---

> > > ### Author Response · Authors · 2018-11-14
> > > **Re: Response to AnonReviewer4 2/3**
> > >
> > > Thank you for taking the time to clarify your question.
> > >
> > > 4) This statement is incorrect: "the prune graph is only trained on data collected when the other agent is present, correct?". The pruned-graph estimator and the full-graph estimator are *both* trained on *all* data. The pruned graph just never gets to know the state of the other agent when making its predictions.
> > >
> > > There’s perhaps a subtlety here, in case you missed it before: all agents are always present in this environment; the difference between the full-graph estimator and the pruned-graph estimator is that one is given access to the state s_a2, and the other is not. In the John/Mary case, the state s_a2 corresponds to the presence/absence of Mary. So your statement that the prune graph “seems to directly model E[X | Mary=1]” is incorrect: it’s directly modelling p(X), where Mary could be 0 or 1 (but it never gets to know what the value truly is).
> > >
> > > As more detail, both the pruned-graph estimator and the full-graph estimator are produced by a single graph neural network. Thus M is the same model in the two equations. We only have one neural network, which is trained to predict agent 1's return both using the full graph (i.e. knowing the actual state of a2) and the pruned graph (i.e. not knowing the actual state of a2). During training we randomly drop out all edges between teammates. At test time, when can then compute the full-graph estimate by using all edges, and the pruned-graph estimator by dropping out edges between teammates.
> > >
> > > We provide this information in the paragraph beginning “We ran this experiment…”. However, we realize from your responses that we did not communicate this clearly, so we have re-written this section by including the following:
> > >
> > > We note that within this setup, both the pruned-graph estimator and the full-graph estimator are produced by a single graph neural network. This network is trained to predict agent 1's return both using the full graph (i.e.\ knowing the actual state of $a_2$) and the pruned graph (i.e.\ not knowing the actual state of $a_2$). During training we randomly drop out edges between teammates (to ensure that both full graph and pruned graph are in-distribution for $M$). At test time, we then compute the full-graph estimate by using all edges, and the pruned-graph estimator by dropping out edges between teammates.
> > >
> > > new question) In the pruned graph, there are no indirect paths between the two agents through the graph. In other words, there are no ways of sending information from s_a2 (from the current input or in the past) through other entities to the node corresponding to a1. There does remain the possibility that the network could infer *something* about s_a2 from the environment state (which we denote z in the main text). For example, if an apple at a particular location was consumed 5 time steps ago, then the network could effectively determine that agent a2 was within a 5-step radius of that location. We make this comment in the footnote at the bottom of pg 8.

---

> > > > ### Comment · AnonReviewer4 · 2018-11-15
> > > > **Re: Response to AnonReviewer4 2/3**
> > > >
> > > > I'll try to be more direct.
> > > > "it’s directly modelling p(X), where Mary could be 0 or 1 (but it never gets to know what the value truly is)."
> > > > Simply because a random variable is unknown doesn't make it 0 or 1 with probability 0.5.
> > > >
> > > > Put another way: Let "Y" be the random variable that's 1 if the other agent is there and 0 otherwise. As you said, "all agents are always present in this environment." Therefore, Y is always set to 1. Simply because Y is unobserved doesn't make Y either 0 or 1 with random probability. If Prune Graph was actually modelling E[X], then you would need to train Prune Graph on data where Y = 0 or Y = 1. However, you always train Prune Graph on data where Y = 1 (because "all agents are always present"). Therefore, Prune Graph models E[X | Y = 1]. Ignoring inputs doesn't change the input distribution; it results in modeling the marginal distribution (with the ignored input being marginalized out). I'm not sure how else to say this, but maybe there is still something that I am missing...
> > > >
> > > > Frankly, given that most of my other concerns have been addressed, I'm inclined to raise my score if Section 2.2.3 is completely removed, or I realize that I am completely mistaken. As it stands, this section seems to be wrong: the method does not seem to be doing what it's claiming. Prune graph models E[X | Y= 1], and not E[X], so one cannot compare E[X | Y=1] with E[X].
> > > >
> > > > I also fail to see why removing some edges from a graph network (a modeling/architecture choice) is equivalent to ignoring the input. The RFM still receives s_a2 as input. (I do not think this concern has been addressed by the authors.)
> > > >
> > > > I cannot, in good conscience, change my rating to an "accept" (even marginally) for a paper with a section that I feel is technically incorrect.

---

> > > > > ### Author Response · Authors · 2018-11-16
> > > > > **Re: Response to AnonReviewer4 2/3 2/2**
> > > > >
> > > > > Q) I also fail to see why removing some edges [...]
> > > > >
> > > > > The RFM indeed receives both s_a1 and s_a2 as input, but it’s computing both R_a1 and R_a2 at the same time. The signal path for the pruned-graph estimator is such that s_a1 information is only routed to predict R_a1, and s_a2 information is only routed to predict R_a2.
> > > > >
> > > > > This can be checked by looking at the Graph Net formulas in Eq. 1 in the paper. Suppose, for simplicity, that there are only 3 vertices:
> > > > >
> > > > > A1 (agent 1) with attributes (x, y, action, N/A)
> > > > > A2 (agent 2) with attributes (x, y, action, N/A)
> > > > > S1 (stag 1) with attributes (x, y, N/A, available/unavailable),
> > > > >
> > > > > 2 edges:
> > > > > E_1: receiver: A1, sender A2, no attributes
> > > > > E_2: receiver: A1, sender S1, no attributes,
> > > > >
> > > > > and that globals are empty.
> > > > >
> > > > > We want to use our graph net to predict the return of agent 1 denoted as A_1’ (to highlight that this will be an updated node attribute). From Eq. 1 in the paper:
> > > > >
> > > > > E_1’ = PHI_E(A_1, A_2)
> > > > > E_2’ = PHI_E(A_1, S_1)
> > > > > A_1’ = PHI_V[PHI_E(A_1, A_2) + PHI_E(A_1, S_1)]
> > > > >
> > > > > (In the last line we used the fact that that RHO_E--->V is just a sum over the senders A_2 and S_1 of the PHI_Es).
> > > > > If we remove edge 1 (receiver A_1, sender A_2) then
> > > > >
> > > > > A_1’ = PHI_V[PHI_E(A_1, S_1)]
> > > > >
> > > > > And S_2 does not enter the calculations.
> > > > >
> > > > > We have also fixed some confusing notation in Eq. 2 and 3 to highlight that M is a function approximator rather than a probability distribution.

---

> > > > > > ### Comment · AnonReviewer4 · 2018-11-16
> > > > > > **Re: Re: Response to AnonReviewer4 2/3 2/2**
> > > > > >
> > > > > > My concern is that there "static entities (i.e., apples, stags,
> > > > > > coins, and tiles) were represented by vertices" and "Edges connected all non-agent entities to all agents as well as agents to each other." So unless I misunderstood, there will still be edges from the stag to a2 and vice versa (and not to mentioned from agent 2 to all the other static entities). Won't this make the pruned graph dependent on sa2 in a non-trivial way?

---

> > > > > ### Author Response · Authors · 2018-11-16
> > > > > **Re: Response to AnonReviewer4 2/3 1/2**
> > > > >
> > > > > Hi, thank you for getting back to us so quickly and for working with us to make sure we put our work out there in a timely fashion.
> > > > >
> > > > > Maybe we finally understand where the confusion comes from: our choice of example with Mary and John and their presence had spurious consequences and we sincerely apologize for this. The variable Y in that example corresponds to Mary’s presence/absence, whereas the variable Y in the paper itself corresponds to the *state* of the agent (e.g. its position, previous action, ...). The fact that you deduce from “all agents are always present in this environment” the conclusion that “Y is always set to 1” (which is not actually the case) might stem directly from this specific choice of example. For the sake of avoiding further misleading statements, let us ground the discussion back in the game we considered.
> > > > >
> > > > > We hope these observations will bring things back into focus:
> > > > > 1. The “state of the teammate”, s_a2, does not denote its presence or absence, but rather all the agent node attributes (position and last action). When we remove s_a2 from the equation we do not “remove the agent”, but simply make information about its position and last action unavailable to the model, M. In light of this, “averaging over Y” makes sense even when “the agent is always there”, because Y contains the agent’s position and last action, rather than an indicator variable denoting its presence.
> > > > >
> > > > > 2. During training, our model has access to examples from all situations (both when using the full graph and the pruned graph). However, when computing the average “Effect on Return” in Fig. 4, we restrict ourselves to situations when a stag was *consumed*.
> > > > > In this case the model with the full graph knows that both agents are on the stag (or near to it, in the time steps leading to the consumption event); the full graph model correctly predicts that both agents will collect a reward of 10. On the other hand the model with the pruned graph does not *know for sure* that the second agent is near or on the stag, so it predicts a lower reward. We measure the difference between these two estimates (one with observed s_a2 and one averaging over an implicit posterior on s_a2) and call it the “value of the actual social context”.
> > > > >
> > > > > 3.You are absolutely correct that if we were to average over *all* situations we would find that the difference between the two estimators is close to 0 (in practice, we find that this difference is less than 5% of the total reward collected by the agents). However, in the analysis of this section, we are not averaging over all situations. Instead, we average only over a specific set of situations: for Fig. 4 middle and right panel, we only consider times when a stag is about to be *consumed*.

---

> > > > > > ### Comment · AnonReviewer4 · 2018-11-16
> > > > > > **Re: Re: Response to AnonReviewer4 2/3 1/2**
> > > > > >
> > > > > > I see. So it seems that in this case, "Y" represents whether or not the stag is about to be consumed. I believe this clarifies the misunderstanding.
> > > > > >
> > > > > > This seems like a very hand-picked event, but it make sense to manually choose events like this to validate the hypothesis. It would be interesting to reverse-engineering this metric to develop a method: use this changes in activation to automatically find interesting events. This could be useful, for example, if you wanted to perform "multi-agent policy distillation." One could use changes in magnitude for attention mechanisms, for supervising intermediate layers, or for prioritized experience replay.
> > > > > >
> > > > > > After this discussion, it is more clear to me how the ideas of this paper can be useful (e.g. for ideas like the ones listed in the previous paragraph). I have updated my score accordingly. I feel that this "collaboration metric proposal" is really the main contribution of the paper, and I would suggest emphasizing that more in the abstract and the last paragraph of the introduction. I suspect a number of first-time readers would still find the language vague and the contribution unclear. However, at this point, these are more suggestions than "reviewer critiques."

---

> ### Author Response · Authors · 2018-11-13
> **Response to AnonReviewer4 1/3**
>
> Thank you for a very thorough and thoughtful review.
>
> 0)  The architecture is a rather straightforward extensions of previous work, and using graph networks for predictive modeling in multi-agent settings has been examined in the past, making the technical contributions not particularly novel. Examining the correlation between edge activation magnitudes and certain events is intriguing and perhaps the most novel aspect of this paper, but it is not clear how or why this information would be useful. There a few unsubstantiated claims that are concerning. There are also some odd experimental decisions and results that should be addressed.
>
> We agree that components of the model are drawn from previous work, and the improvements to this part of the architecture are incremental. However, our two major novel contributions are elsewhere. The first, as the reviewer points out, is in using this model for analysis of multi-agent behavior. This is useful for teasing apart patterns of influence in complex situations; for instance, we can use this method to answer AnonReviewer3’s questions about whether (and when) agents are coordinating with each other. Such analysis can also assist with the evaluation of different multi-agent algorithms to determine whether they are producing desirable policies, or to more finely dissect the cooperative or competitive behaviors that a task induces.
>
> Our second contribution is to integrate the RFM model into agents, which we show assists with their decision-making. This provides a measurable improvement over baselines in multi-agent tasks.
>
>
> 1)  Why would using a recurrent network help (i.e. RFM vs Feedforward)? Unless the policies are non-Markovian, the entire prediction problem should Markovian. I suspect that most of the gains are coming from the fact that the RFM method simply has more parameters than the Feedforward method (e.g. it can amortize some of the computation into the recurrent part of the network). Suggestion: train a Feedforward model that has more parameters (with appropriate hyperparameter sweeps) to see if this is the cause. If not, provide some analysis for why “memories of the relations between entities” would be any more beneficial than simply recomputing those relations.
>
> By the nature of the problem, we do expect a priori that a stateful RFM should outperform a stateless one, for two reasons. First, in all cases, the agents that the RFMs were modelling were themselves stateful. If the agents are making any functional use of their memory (which we anticipate in the general case), then the RFM would benefit from taking advantage of previous relations between entities. Second, while CoopNav and StagHunt are fully observed, CoinGame is not, as the episode-specific reward for each agent’s coins are known to the agents themselves, but not to their teammates (or the RFM). This latent variable has to be inferred from teammates’ history of actions, since its value is often aliased within a single observation. From the results in Figure 2, we indeed find that the recurrent RFM outperforms the feedforward RFM the most when modelling behavior in CoinGame.
>
>
> 2)  The other potential reason that the recurrent method did better is that policy actions are highly correlated (e.g. because agents move in straight lines to locations). If so, then recurrent methods can outperform feedforward methods without having to learn anything about what actually causes policies to move in certain directions. Suggestion: measure the correlation between consecutive actions. If there is non-trivial correlation, than this suggests that RFM does better than Feedforward (which is basically prior work of Battaglia et. al.) is for the wrong reasons.
>
> We don’t think this alternative hypothesis explains the improved performance of the recurrent model, for two reasons. First, the feedforward RFM includes the most recent previous action in its inputs, allowing it to take advantage of correlations between consecutive actions. Second, any autocorrelation structure in an action sequence is either a consequence of autocorrelation in the MDP state (in which case a feedforward RFM should be able to reproduce it), or it is due to statefulness of the agent (in which case a recurrent RFM is necessary; see our argument above). Either which way, the RFM model will indeed learn what actually causes policies to move in certain directions.

---

> > ### Comment · AnonReviewer4 · 2018-11-13
> > **Re: Response to AnonReviewer4 1/3**
> >
> > 1) I did not realize that the policies were stateful and forgot that the coin game is partially observed. That would also explain why there is a larger difference in that environment. Thank you for this clarification.
> > 2) Great point!
> >
> > This was a smaller concern, but are there any comments regarding the fact that the recurrent model simply has more parameters than Feedforward?

---

> > > ### Author Response · Authors · 2018-11-14
> > > **Re: Response to AnonReviewer4 1/3**
> > >
> > > Thank you for your question!
> > >
> > > We tried to match models for capacity (with the exception of NRI which has about 3x more parameters than other models because of its autoencoder connectivity estimator). The raw number of parameters (as reported by the TensorFlow checkpoint loader) were as follows for the RFM, FeedForward and MLP + LSTM:
> > >
> > > (RFM, CoopNav) ---> 61194
> > > (RFM, CoinGame) ---> 63134
> > > (RFM, StagHunt) ---> 63134
> > >
> > > (FeedForward, CoopNav) ---> 60240
> > > (FeedForward, CoinGame) ---> 65140
> > > (FeedForward, StagHunt) ---> 65140
> > >
> > > (MLP + LSTM, CoopNav) ---> 59307
> > > (MLP + LSTM, CoopNav) ---> 66718
> > > (MLP + LSTM, StagHunt) ---> 71803
> > >
> > > We don’t think a discrepancy in the number of parameters of less than 3% can account for the difference in performance we observe.
> > >
> > > We have added a comment to the paper (“We matched all models for capacity…”).

---

### Public Comment · (anonymous) · 2018-12-21
**Great work! A couple of missing references**

Enjoyed reading this work and congrats on the acceptance! Also wanted to bring your attention to a pair of papers that  use relational interactions between agents in cooperative and competitive environments to accelerate multi-agent learning that you might find relevant:

Evaluating Generalization in Multiagent Systems using Agent-Interaction Graphs
Aditya Grover, Maruan Al-Shedivat, Jayesh K. Gupta, Yura Burda, Harrison Edwards
AAMAS 2018 (short paper)
Link: https://dl.acm.org/citation.cfm?id=3238032

Learning Policy Representations in Multiagent Systems
Aditya Grover, Maruan Al-Shedivat, Jayesh K. Gupta, Yura Burda, Harrison Edwards
ICML 2018
Link: https://arxiv.org/abs/1806.06464

---

### Meta-Review · Area_Chair1 · 2018-12-12
**Good paper showing the benefit of relational representations in a multi-agent setting**

**Confidence:** 3
**Recommendation:** Accept (Poster)

**Metareview:**


pros:
- interesting application of graph networks for relational inference in MARL, allowing interpretability and, as the results show, increasing performance
- better learning curves in several games
- somewhat better forward prediction than baselines

cons:
- perhaps some lingering confusion about the amount of improvement over the LSTM+MLP baseline

Many of the reviewer's other issues have been addressed in revision and I recommend acceptance.